# Cognitive assessment after stroke: A qualitative study of patients' experiences

Georgina Hobden ![ORCID],[1] Eugene Tang,[2] Nele Demeyere[3]

¹Experimental Psychology, University of Oxford, Oxford, UK
²Population Health Sciences Institute, Newcastle University Faculty of Medical Sciences, Newcastle upon Tyne, UK
³Nuffield Department of Clinical Neurosciences, University of Oxford, Oxford, UK

**Correspondence to**
Georgina Hobden;
georgina.hobden@psy.ox.ac.uk

## ABSTRACT

**Objectives** Clinical guidelines recommend early cognitive assessment after stroke to inform rehabilitation and discharge decisions. However, little is known about stroke survivors' experiences of the cognitive assessment process. This qualitative study aimed to explore patients' experiences of poststroke cognitive assessments.

**Design** Stroke survivors were purposively sampled in an iterative process through a pool of research volunteers who had previously taken part in the Oxford Cognitive Screen Recovery study. Stroke survivors and their family caregivers were invited to participate in a semistructured interview steered by a topic guide. Interviews were audio recorded, transcribed and analysed using reflexive thematic analysis. Demographic, clinical and cognitive data were acquired from patients' previous research data.

**Setting** Stroke survivors were originally recruited from the acute inpatient unit at Oxford University Hospital (John Radcliffe), UK. Participants were interviewed after discharge either at their homes or via telephone or videocall.

**Participants** Twenty-six stroke survivors and eleven caregivers participated in semi-structured interviews.

**Results** We identified three key phases of the cognitive assessment process and themes pertaining to each phase. The phases (numbered) and themes (lettered) were as follows: (1) before the cognitive assessment: (A) lack of explanation, (B) considering the assessment useless; (2) during the cognitive assessment: varied emotional responses, moderated by (D) perception of the purpose behind cognitive assessment, (E) perception of cognitive impairment, (F) confidence in cognitive abilities, (G) assessment administration style and (3) after the cognitive assessment: (H) feedback can impact self-confidence and self-efficacy, (I) vague feedback and clinical jargon are unhelpful.

**Conclusions** Stroke survivors require clear explanations about the purpose and outcomes of poststroke cognitive assessments, including constructive feedback, to promote engagement with the process and protect their psychological wellbeing.

## INTRODUCTION

Stroke is among the most common causes of disability worldwide and nearly all patients experience some level of cognitive impairment in the first weeks after stroke.[1–3] Acute cognitive impairment has implications for poststroke recovery by increasing the risk of poststroke depression[4 5] and reducing quality of life.[6]

Given the high prevalence of poststroke cognitive impairment and its implications for recovery, national and international clinical guidelines recommend that cognitive functioning should be assessed as soon as possible after stroke (e.g. National Institute for Health and Care Excellence guideline for stroke care, 2013); Royal College of Physicians clinical guideline for stroke).[7] Cognitive assessment tools designed specifically for stroke populations (e.g. Oxford Cognitive Screen, OCS)[8] are increasingly used for this purpose, as well as in community stroke settings.[9] Stroke-specific tools, such as the OCS, are usually administered by occupational therapists as a form of first-line screening for cognitive problems after stroke and scores from these assessments are used to inform and plan rehabilitation programmes,[9] which may include further referral to clinical neuropsychology services where appropriate.

Nevertheless, very little is known about stroke survivors' experiences of the cognitive assessment process. This is despite increasing recognition of the value of patient-centred outcome research, which places emphasis on understanding patients' experiences, preferences and needs to improve clinical practice.[10 11] Patient-centred outcome research in non-stroke populations (e.g. geriatric inpatients,[12] people with multiple sclerosis[13]) indicates that some patients may not fully understand the purpose of cognitive assessments and that cognitive assessments have the potential to provoke intense emotions, eliciting feelings of shame and irritation among some patients.[12 13] It is critical to explore whether stroke survivors similarly lack understanding about the purpose of stroke-specific cognitive assessments and whether the poststroke cognitive assessment process also elicits negative emotional responses, in order to identify ways to protect stroke survivors' psychological and emotional wellbeing throughout this process.

However, to our knowledge, stroke survivors' experiences, preferences and needs surrounding the poststroke cognitive assessment process have not yet been investigated. The purpose of this study was to explore experiences of cognitive assessment after stroke using semistructured interviews with stroke survivors and their caregivers.

## METHODS
### Patient and public involvement
Patients were involved in the development of the fellowship funding proposal of which this study forms a part. Patients were consulted on the importance of the question and the methodological approach through a survey with the Stroke Association's Voices in Research (*n* = 43) and three smaller focus groups. With regard to the present study, patients suggested holding the interviews over videocall, rather than over the telephone, where in-person visits were not possible to enhance rapport between interviewee and interviewer. They stressed the importance of discussions about cognitive changes being conducted sensitively (e.g. avoiding emotive words) and they emphasised the need to include carer experiences where possible.

### Participant sampling
Stroke survivors were recruited through a pool of research volunteers who had previously taken part in the OCS-Recovery study (NHS REC reference: 18/SC/05501). The OCS-Recovery study originally recruited a sample of stroke survivors from the acute stroke inpatient unit at Oxford University Hospital (John Radcliffe), United Kingdom. Stroke survivors recruited to the OCS-Recovery study were visited for a follow-up assessment at their home after discharge. The average (mean) time since stroke at follow-up for the OCS-Recovery participant sample was approximately 6-months. Inclusion criteria for the OCS-Recovery study were: at least 18 years old, clinical diagnosis of stroke and ability to remain alert for at least 20 minutes at the point of recruitment. Patients provided written or witnessed informed consent at recruitment and at follow-up.

The present study iteratively sampled stroke survivors who had already completed their follow-up visit and provided opt-in consent to be contacted about further research participation. The research team considered that stroke survivors with different characteristics, including sex, level of stroke severity and types of domain-specific cognitive impairment, would likely hold different but important views pertaining to the main research question. Therefore, purposive sampling was used to ensure the sample included both male and female stroke survivors, stroke survivors with mild, moderate, and severe stroke, and stroke survivors with domain-specific cognitive impairments affecting different domains. Stroke severity was assessed using the National Institute of Health Stroke Scale (NIHSS) score, which was recorded during acute hospital admission.[14] Cognitive impairment was determined based on performance on the OCS relative to normative cut-offs during acute hospital admission. Family caregivers of sampled stroke survivors were also invited to participate.

GH initially contacted sampled participants by telephone to provide them with a brief description of the study and offer them the opportunity to ask any questions. They were then sent a detailed information sheet and participant consent form to read. They were asked to phone the research team to ask any questions and to organise a date for the semistructured interview, should they decide to participate.

### Procedure
The research team developed a topic guide for the semistructured interviews based on relevant literature[12 13] and their clinical expertise. It was designed to be iterative so that topics not originally identified could be pursued in interviews. The topic guide comprised mostly open questions to allow flexible dialogue and ad hoc questioning, and to encourage participants to share their thoughts without undue biasing. However, this questioning style was adjusted for stroke survivors with aphasia and communication support strategies were used (e.g. probing, word offerings).[15]

The topic guide included questions about: (1) experiences of poststroke cognitive assessment; (2) experiences dealing with cognitive changes after stroke; (3) thoughts about cognitive care after discharge and (4) perceptions of the value of discussing cognitive trajectories after stroke. This paper reports experiences of cognitive assessment only. Given the breadth and depth of the interviews, the other interview topics will be analysed and discussed separately.

GH conducted interviews between May 2022 and September 2022. Semistructured interviews were conducted either at participants' homes or via telephone or videocall, with interview location determined

by participants' individual preferences. Most interviews were conducted individually (i.e. interviewer and one participant). However, they were conducted in pairs (i.e. interviewer and two participants) if both the stroke survivor and their caregiver preferred to take part in the interview together. Interviews were audio recorded with participants' consent. GH transcribed audio recordings verbatim using Jeffersonian Lite style.[16] All identifiable personal data were removed from the transcripts and transcripts were labelled using unique participant identifiers.

Demographic and clinical data were retrieved from previous research data. Clinical data included NIHSS score, a measure of stroke severity recorded by clinical staff during acute hospital admission,[14] and performance on the OCS. The OCS was administered by occupational therapists as part of routine care during hospitalisation for stroke using the procedure outlined by the OCS manual.[8] Cognitive impairment classifications were determined using normative data from neurotypical adults. OCS administration, scoring and classification of impairment are described in detail elsewhere.[8]

### Data analysis

Data analysis was facilitated by NVivo (V.11) and Microsoft Excel. Data were iteratively sampled and analysed using a grounded theory approach and techniques drawn from reflexive thematic analysis.[17 18] The research team adopted a reflexive stance during data collection and analysis, bearing in mind their professional roles and clinical expertise. We used both deductive and inductive approaches, in that we derived new themes inductively from our data, as well as using previous research to inform theme development. The decision to cease data collection and analysis was based on several factors, including the analytic goals of the study (i.e. to develop a theme structure encapsulating experiences of cognitive assessment after stroke) and pragmatic constraints on time and resources.[19]

GH conducted data analysis. After transcribing and coding a subset of interviews ($n = 8$), GH and ET discussed initial codes and themes. As interviews progressed and transcripts were revisited, initial themes were refined, and new themes were developed. These were discussed between GH and ET at regular intervals. All members of the research team discussed and agreed on the final themes.

### RESULTS

Thirty-seven participants were interviewed, including 26 stroke survivors and 11 caregivers. All participants completed one interview session, except one stroke survivor and their caregiver, who completed two interview sessions, as a technical error resulted in deletion of the first audio recording. Seventeen interviews were conducted in person and eleven interviews were conducted remotely via telephone. No interviews were conducted by videocall. Interviews lasted on average 47 min (range = 14–119 minutes).

Table 1 reports demographic and clinical data for the stroke survivors interviewed in the present study and Table 2 reports the relationship of caregivers to stroke survivors.

Participants discussed three main phases of the post-stroke cognitive assessment process and we identified themes pertaining to each phase. Phases were: (1) before the cognitive assessment, (2) during the cognitive assessment and (3) after the cognitive assessment. Themes pertaining to each phase are described below.

**Table 1** Demographic and clinical data for stroke survivors interviewed for the present study

| Variable | Statistics |
|---|---|
| Age range, (M, SD) | 54–87 (72.77, 8.18) |
| Sex, N (%) | Female: 9 (34.62); Male: 17 (65.38) |
| Acute OCS task impairment range, (M, SD) | 0–11 (3.85, 3.11) |
| Acute NIHSS range, (M, SD) | 0–18 (7.17, 5.52) |
| Days since stroke at interview range, (M, SD) | 109–631 (345.46, 187.36) |

Acute OCS refers to performance on the OCS during acute hospital admission. We report the average number of tasks on which stroke survivors were categorised as impaired, relative to normative data, with higher scores denoting poorer performance (i.e. more task impairments). The OCS includes a total of 12 tasks. Acute NIHSS refers to stroke severity. This was assessed during acute hospital admission using the National Institute of Health Stroke Scale, which ranges from 0 to 42.
NIHSS, National Institute of Health Stroke Scale; OCS, Oxford Cognitive Screen.

**Table 2** Relationships of caregivers Interviewed for the present study to stroke survivors

| Unique ID | Role | Relationship to stroke survivor |
|---|---|---|
| C1 | Carer of P5 | Wife |
| C2 | Carer of P6 | Husband |
| C3 | Carer of P7 | Wife |
| C4 | Carer of P9 | Wife |
| C5 | Carer (stroke survivor not interviewed) | Daughter |
| C6 | Carer of P15 | Wife |
| C7 | Carer of P17 | Wife |
| C8 | Carer (stroke survivor not interviewed) | Son |
| C9 | Carer of P20 | Wife |
| C10 | Carer of P22 | Wife |
| C11 | Carer of P24 | Wife |

## Before the cognitive assessment

### Lack of explanation

Some participants referred to the fact that the assessment administrator 'didn't explain much' (P12) about the cognitive assessment before beginning the tasks. In particular, participants seemed to feel as though the purpose of the assessment was not clearly explained to them.

> P12: I seem to remember this animal—which I thought might have been a hippo and it wasn't—and we discussed what it might be. Then I had to look for the same shaped animal on some other picture and say "There it is! There it is! There it is!" I don't recollect [the assessment administrator saying] anything other than this is just seeing what your brain's like.

Some participants who claimed they received little explanation about the cognitive assessment and its purpose spoke as though they were simply doing the assessment administrator a favour. This was particularly true of stroke survivors without cognitive impairment, raising the possibility that their intact cognitive abilities led the assessment administrator to skim over the explanation.

> P7: [The cognitive assessments] were alright, you know. I persevered with 'em. Yeah, okay, you want me to do that, I'll do it. I don't really want to.

Stroke survivors generally showed a very limited understanding of the purpose behind cognitive assessments, which many participants attributed to the lack of explanation they received beforehand. When asked about their understanding of the purpose behind post-stroke cognitive assessments, some participants failed to refer to cognition or thinking abilities at all. For example, one participant stated that cognitive assessments are used 'just [to see] what your brain's like' (P12). Others showed a very basic understanding of how cognitive assessments are used to detect problems in patients' thinking abilities after stroke. However, participants tended to describe this in vague and sometimes circular terms.

> P1: Basically, it's an assessment to see how you are and… I don't suppose they used the phrase 'what damage has been done' but to get an assessment of where you're at, what your cognitive state is really.

In several interviews, participants referred to cognitive assessments as 'memory tests' (P6, P13, C4, C5). These participants seemed to recognise the role of poststroke cognitive assessments in detecting memory problems, but not deficits in other cognitive domains. No participants mentioned the role of poststroke cognitive assessments in detecting aphasia or visuospatial neglect, despite several participants having demonstrated these domain-specific cognitive impairments on cognitive screening tests.

> C5: All she's had is a memory test. Nobody else, to my knowledge, has done anything with her.

### Considering the assessment useless

The lack of explanation about the cognitive assessment meant that many participants failed to recognise the value of cognitive assessments—for example, for shaping their rehabilitation programmes and care packages. Some participants employed pejorative expressions to describe them, such as 'stupid' (C8), 'pointless' (P14) and 'a load of old c***' (P5).

> C8: She was like "Why am I doing these stupid things?" and all that.

Viewing the cognitive assessments as useless was particularly common among participants who presented with cognitive impairments on the OCS while in hospital. Viewing the assessment as useless may have served as a psychological defence mechanism to deny the cognitive impact of stroke. One participant with impairments in six tasks on the OCS argued that cognitive assessments are pointless as they are too 'naïve' (P20) to detect cognitive impairment. Another participant with impairments in four tasks and expressive aphasia referred simply to the assessments as 'a load of old c***' (P5) and described how this perspective was a barrier to engaging with the process.

> P20: The questions were naïve in their… You can't do a questionnaire like that when your audience is so varied, the people taking it. I don't think it works.

> P5: I didn't really engage with [the cognitive assessment] anyway because I thought "oh, it's a load of old c***".

## During the cognitive assessment

When stroke survivors discussed their experiences completing the cognitive assessment, the overarching theme was that cognitive assessments provoke a variety of emotional responses. Whereas some participants appeared to enjoy the assessment, describing the experience as 'fun' (P12) and a welcome 'break from the boredom of being in hospital' (P21), other participants reported negative emotional responses to the assessment, including frustration, distress and anxiety. Several factors appeared to moderate these emotional responses to the cognitive assessment process. These factors are described below.

### Perception of the purpose behind cognitive assessment

Stroke survivors' perception of the purpose behind the assessment seemed to influence their emotional response. One participant who reported enjoying the assessment referred to it benignly as 'a little challenge' (P25), whereas several other participants perceived the assessment more threateningly as a 'test' or 'exam', designed to pinpoint weaknesses in their cognitive abilities. Some participants drew comparisons between the cognitive assessment and their experiences of anxiety-provoking exams in childhood, speculating that their anxiety may have impeded their performance.

P4: One thing that's happened here is I'm constantly worrying, just as I did at school, you know, if I got… say in Maths or Arithmetic, as it used to be called, you suddenly had a test in front of you, and I literally found myself thinking "No, no, I can't." And that made seeing it or focusing on it much more difficult.

P5: If I've got a test coming towards me, I'll get "oh Christ" because it gets back to school, back to tests kind of thing, so I'll get that…anxiousness…anxiety about it.

Perceiving the assessment as a 'test' or 'exam' tended to be 'kind of motivating but motivating in a negative way' (P5) for stroke survivors, as they felt they had to prove something to themselves or to the assessment administrator.

C7: You [P17] were quite anxious beforehand about the assessment. You appeared very calm, but your concentration was ferocious in doing it. Almost I could see sort of a bubble coming out of your head saying "I'm b***** well going to show them and I'm going to get all these answers right."

This feeling seemed to interact with patients' diminished self-confidence after stroke, as some stroke survivors described how performing poorly in the 'test' made them feel 'not good enough' (P5). Others described more broadly their fear that the assessment would reflect their self-worth.

P4: Oh my god! I'm…I'm being tested here. The result will reflect what I am.

### Perception of cognitive impairment

Stroke survivors' emotional responses to the cognitive assessment were also shaped by their perception of post-stroke cognitive impairment. Recognising the potential to recover from cognitive impairment after stroke meant that patients were less likely to endorse cognitive impairment as a shameful element of their personal identity, and instead saw it as something that could be overcome with time. In turn, this reduced feelings of anxiety about the cognitive assessment itself. Similarly, by realising that stroke and cognitive impairment can affect anyone, stroke survivors were protected from feelings of shame and embarrassment while completing the tasks, as they identified cognitive impairment as a potential part of the 'human condition' (P20), rather than as a problem exclusive to them.

P20: I wasn't particularly concerned [when I got a question wrong]. I probably didn't realise the implications of it, and I just wanted to get onto the next part of the questionnaire or whatever it was we were engaged in. And I wasn't in any way embarrassed by it and that's actually been a part of the stroke all the way through. I mean, 'lots of people have strokes' was my sort of global rationale for this terrible thing that has happened to me. Lots of people have strokes and a lot of them get over these strokes, so that kept me going, even if I didn't know the answers to the next question. And I've never been…I've not been at all embarrassed by this. I just see it as part of the human condition.

Similarly, several participants with cognitive impairments adopted a pragmatic problem-solving perspective, viewing poststroke cognitive impairment as an issue to be addressed, rather than a stigmatising diagnostic label.

P16: If you asked anybody who knew me, they would say, you know, my glass is definitely…it's not half full, it's three quarters full! I'm very positive about everything and if things aren't quite right, well, then you either change them or you make them right somehow.

P20: My way through this at that point was we have to fix this, and we have to come back out and get over the stroke, and I'll use any method I can to do that, so there was no soppiness.

### Confidence in cognitive abilities

Emotional responses to the cognitive assessment were also moderated by patients' confidence in their own cognitive abilities after stroke. Patients who were confident that their stroke had not affected their cognitive skills tended to be less anxious during the cognitive assessment, compared with stroke survivors who expected the assessment to detect weaknesses in their cognitive abilities.

P3: Maybe if I had cognitive problems from the stroke, I might have felt different and more awkward about [the cognitive assessment].

P5: I feel actually quite nervous because I think "I'm not going to get this right, I'm going to get it wrong".

Similarly, participants' emotional responses during the assessment often reflected their perceptions of their performance in the assessment. Whereas stroke survivors described feelings of 'relief' (P24) when they felt they had done well in the assessment, participants who felt they had done poorly described feelings of failure or shame.

P17: I felt that I was failing it, you know, so it felt pretty lousy, to tell the truth. It felt pretty lousy.

### Assessment administration style

Finally, patients' emotional responses during the cognitive assessment were influenced by external factors, such as how the cognitive assessment was administered. Several stroke survivors expressed gratitude for the 'gentleness' (P4) and patience with which the therapist administered their cognitive assessment, as this approach made them feel more comfortable during the assessment and reduced feelings of anxiety. One stroke survivor highlighted the brutality of the hospital environment and explained how cognitive assessments—when administered with patience and gentleness—could provide an important counterpoint.

P4: Really the thing about hospitals is how brutal they have to be in some ways and the thing that I, you know, once or twice thought about was but please be gentle too because we're lying here, not quite able to explain to you how we're feeling.

In direct contrast, one participant with aphasia described the negative emotional consequences of pressurised administration, explaining how perceived pressure to answer questions quickly compounded his frustration during the cognitive assessment.

P17: I was so frustration…frustration…frustrated that I couldn't actually do this. A! B! I also felt under tremendous pressure to get answers quickly, kept going, and I found that quite tricky that, you know, I should be…I felt I should be doing twenty answers, for example, but would do only three, so I was very frustrated. There's…I felt a sense of pressure from the therapists that I needed to things and get on and sort things. And…so…I didn't…Not only I was frustrated but I was also…I felt that I was failing it.

Similarly, several caregivers felt that the assessment administrator did not sufficiently account for the impact of poststoke fatigue, pushing patients to continue with the assessment when they were tired, which affected both patients' test performance and emotional response during the assessment.

C7: Sometimes the therapist didn't realise just how tired you feel after you've had a stroke and they would come in bright and breezy, all cheery, which was lovely, but they didn't always seem to know when that was enough and [P17] was actually getting very tired and therefore not getting so many answers right, which increased the frustration and the sense of him feeling that he was a failure because he couldn't get them.

### After the cognitive assessment

Stroke survivors reported receiving varying amounts and types of feedback after completing the cognitive assessments, suggesting a potential lack of consistency in the reporting of cognitive assessment results. Several participants described the encouraging effects of positive feedback (i.e. feedback highlighting intact cognitive abilities), whereas others referred to the feedback as vague and unhelpfully complicated by jargon.

### Feedback can impact self-confidence and self-efficacy

Several participants drew a contrast between the confidence-damaging nature of stroke and the potential for positive feedback to restore, or at least increase, diminished self-confidence. This was particularly common among participants with cognitive impairments, suggesting that encouragement is especially important when patients are struggling with the tasks. Participants described how positive feedback after completing the cognitive assessment provided much needed reassurance

during a period of vulnerability that had arrived without warning or expectation.

P4: For many of us, I guess we temporarily are, we're back in our childhood, where to get approval and to have to somebody say "Well, that is, you know, really amazingly good!", even if it isn't, will make the world of difference, I think, because it brings back the confidence, which has been absolutely shattered by this blow from above.

P1: It's quite an experience having a stroke. I was really knocked for six because I didn't think I was in that category. […] And to be able to complete well the tests, I was quite pleased really. […] The person that came round was very reassuring that you were doing really well so that made me feel quite good.

Many stroke survivors adopted a critical perspective of their own performance in the cognitive assessment, assuming they had done poorly. These participants appreciated positive feedback, as oftentimes it reassured them that their skills were intact and promoted feelings of self-efficacy (i.e. confidence in their skills/abilities).

P18: At first, when I completed the [cognitive assessment], I was a bit maybe disappointed, thinking "Oh, I didn't do very well on this. […]" But when [the therapist] explained to me that this is absolutely quite normal and in fact I was at least average, if not better than average given my particular situation, it made me feel a lot…a lot…okay, and making decent progress, so it was quite positive.

One caregiver referred to her husband's self-critical tendencies after the stroke and speculated that professional feedback highlighting his cognitive strengths could have reminded him of his intact abilities and increase his self-efficacy, but she was disappointed this had not happened in practice. Instead, she described how the small amount of feedback provided was biased towards the patient's weaknesses, rather than encapsulating areas in which he had performed well.

C1: Possibly [P5] is his own worst critic…and sometimes he doesn't criticise him enough! But he has me then! (Laughs) But what I mean is that some objective outside discussion would be very, very helpful.

C1: I mean, they've been terribly good at telling us how bad [P5] is but they're not terribly good at telling us how good he is.

### Vague feedback and clinical jargon are unhelpful

Some stroke survivors and caregivers felt disappointed at the lack of concrete feedback provided after their cognitive assessment. Several participants described how the assessment administrator wrote down results, without communicating them directly. One participant reflected that such behaviour could leave patients with 'the vague feeling 'Oh dear, I made a right mess." (P4), which she suggested could be damaging to patients' self-confidence.

C1: There's a lot of stuff written but then that's not discussed with you. And that's actually…that aspect is not good.

Several participants also discussed how the use of complex terminology and clinical jargon was a barrier to understanding feedback from the cognitive assessments, highlighting the need for feedback to be communicated to patients clearly using lay language.

P17: (Reading from therapist report) "I get 13 out of 16. Not bad. Visuospatial." Well, I don't b***** hell what that means! 'Visuospatial'!

Some participants felt so alienated and confused by clinical jargon that they resorted to seeking definitions themselves. One participant described how he consulted the dictionary to understand the meaning of the word 'cognition'.

P17: I went out to the […] dictionary about three months ago to ask one question—"What does that mean?" […] 'Cognition' […] (Reading from a typed sheet of paper) 'What are the eight cognitive skills? The eight core cognitive capacities. Sustained attention… Response…' Well, that's very helpful! I thought that was actually very useful, thank you very much, Mr Wikipedia or whoever you are! That's exactly what I wanted. I wanted to know what cognition meant.

Some participants provided suggestions to improve the interpretability of feedback after cognitive assessments. For example, one participant suggested that visual feedback might be better than verbal feedback as this would avoid clinical jargon. He suggested that a visual representation would be particularly helpful if a cognitive assessment was repeated in order to help the patient visualise any progress they had made.

C8: Something visual is probably the easiest. Jargon-y words are probably not so helpful and a score that just says "You got 10 the first time and 40 the second", I don't think is that much help, but anything that is visual would be quite encouraging, I think.

Another participant suggested linking cognitive impairments to activities of daily living when providing feedback. She reflected that illustrating the impact of cognitive impairments on daily life would have made feedback more interpretable.

C7: You've always enjoyed cooking. It's got a bit better now but actually, if there's a menu, you have great difficulty trying transferring from reading—not great difficulty—but reading what it says and then sort of working out what you therefore do or what order. And again, that's a sort of tiny bit of one of cognition, probably one of those elements. But actually, somebody saying "Well, let's illustrate this with how you go about cooking" would be really helpful, I think. Yup. I think big words don't help.

## DISCUSSION

The present study used a qualitative approach to explore experiences of the cognitive assessment process from the perspective of stroke survivors and their family members. While recommended by clinical guidelines, poststroke cognitive assessments had the potential to evoke strong emotional and psychological responses, which should be handled carefully by healthcare professionals. The present study highlights the potential benefit of providing a clear explanation of the purpose behind the cognitive assessment and constructive feedback afterwards. The development of clear guidelines and recommendations may help to standardise this process.

Stroke survivors in the present study generally showed a very limited understanding of the purpose behind poststroke cognitive assessments and their role in guiding clinical care, which served as a barrier to engaging with the process. While this echoes the results of previous studies in other clinical populations,[12 13] it was not entirely clear why stroke survivors in the present study failed to understand the purpose behind poststroke cognitive assessment. Although some participants put their limited understanding down to the lack of explanation provided by the assessment administrator, it is also possible that they may have struggled to interpret, understand and/or retain the information provided. Future research should determine how healthcare professionals might best communicate the purpose and significance of the poststroke cognitive assessment process, and how they can be confident the information is understood. This is important as our interviews revealed that participants who did not understand the purpose of cognitive assessments sometimes failed to engage with them, which may limit the validity and accuracy of cognitive assessments conducted in clinical practice.

Cognitive assessments triggered various emotional responses, which were moderated by several different factors. First, perceiving the assessment as a test triggered feelings of anxiety, in line with the phenomenon of test anxiety.[20] Crucially, not only is test anxiety potentially detrimental to emotional wellbeing, it may also disrupt attention and memory function, potentially impacting the validity of assessment results.[21] It is worth considering, therefore, whether patient-facing language should be adapted so that terms such as 'puzzles' or 'activities' replace the anxiety-provoking clinical terminology of 'tests', 'examinations' and 'screens'. Second, recognising the potential to recover from poststroke cognitive impairment protected patients from feeling shame and embarrassment, which suggests that they may benefit from healthcare professionals outlining stroke recovery statistics for specific cognitive domain impairments[2 22] during the assessment process. Finally, perceived pressure during assessment administration was detrimental to psychological wellbeing, highlighting the need for healthcare professionals to set aside sufficient time to complete cognitive screening, and to be mindful of being supportive and constructive during the assessment.

Stroke survivors reported receiving different amounts and types of feedback after the cognitive assessment, indicating a potential lack of consistency in the feedback provided in clinical practice. This highlights a need for clearer guidelines and tools to standardise what and how information is communicated to stroke survivors and their caregivers after cognitive assessments. Future recommendations should consider the emotional impact of cognitive assessment feedback, as the present study revealed the potential for feedback to improve confidence and self-efficacy, both of which have been identified as important factors in promoting poststroke recovery.[23 24] Our interviews also demonstrated the importance of communicating feedback using lay terminology, echoing the results of previous research that looked at patient responses to jargon in other clinical populations.[25] However, further research is required to clarify precisely which words hinder patients' understanding and to identify acceptable alternatives.

We note several limitations of the present study. First, participants were interviewed several months after their stroke and poststroke cognitive assessment, so their memory of the process may have faded. Nevertheless, participants often reported intense emotional responses to the assessment, suggesting that it was a salient and memorable experience for many. A second related limitation is that we cannot verify the information provided to stroke survivors during the cognitive assessment process. A particular concern is whether stroke survivors were given veridical feedback that accurately reflected their performance in the assessments, whether they received positively skewed general feedback that failed to mention areas of impaired cognition, or whether in fact they received no feedback. According to their research records, most stroke survivors who took part in the present study showed impairments in at least one task in the cognitive assessment administered during acute hospitalisation, but very few participants reported receiving feedback about their cognitive impairments. As only a few participants reported receiving feedback on their impairments, a related limitation is that our interviews provide only limited insight into stroke survivors' responses to specific cognitive domain impairment-focused feedback after poststroke cognitive assessment.

The present study demonstrates the importance of communicating clearly, carefully, and constructively throughout the poststroke cognitive assessment process, to promote patients' engagement with cognitive assessments and protect their psychological and emotional wellbeing. Our study also highlights the need for clearer guidelines and clinical tools to standardise and promote information provision about the purpose and results of cognitive assessments after stroke.

**Acknowledgements** We would like to thank the participants who took part in the study. We would also like to thank the research staff who contributed towards data collection for the original OCS-Recovery programme. In particular, we acknowledge the contributions to data collection and curation for the OCS data made by Mr Sam Webb, Dr Margaret Moore and Ms Evangeline Grace Chiu.

**Contributors** GH conceived the framework for this study. GH collected, transcribed, analysed and interpreted the data. GH prepared the manuscript for submission. ET assisted with the data analysis and drafting of the manuscript. ND helped conceive the framework for this study, assisted with the analysis of the data and critically reviewed and edited the manuscript. ND accepts full responsibility for this work.

**Funding** GH is supported by an Economic and Social Research Council (ESRC) grant (ES/P000649/1). ET, NIHR Clinical Lecturer, is funded by the National Institute for Health Research (NIHR). ND (Advanced Fellowship NIHR302224) is funded by the National Institute for Health Research (NIHR).

**Disclaimer** The views expressed in this publication are those of the authors and not necessarily those of the ESRC or NIHR, NHS or the UK Department of Health and Social Care.

**Competing interests** None declared.

**Patient and public involvement** Patients and/or the public were involved in the design, or conduct, or reporting, or dissemination plans of this research. Refer to the Methods section for further details.

**Patient consent for publication** Not applicable.

**Ethics approval** This study involves human participants and was approved by University of Oxford Medical Sciences Interdivisional Research Ethics Committee (Ethics Approval Reference: R80681/RE001). Participants gave informed consent to participate in the study before taking part.

**Provenance and peer review** Not commissioned; externally peer reviewed.

**Data availability statement** No data are available. Ethical constraints prevent sharing of interview transcripts to protect participant anonymity.

**ORCID iD**
Georgina Hobden http://orcid.org/0000-0002-4443-296X

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
