## [Reviewer comments · BMJ Open]

ARTICLE DETAILS

TITLE (PROVISIONAL)	Cognitive assessment after stroke: A qualitative study of patients' experiences
AUTHORS	Hobden, Georgina; Tang, Eugene; Demeyere, Nele

VERSION 1 – REVIEW

REVIEWER	Hofgren, Caisa University of Gothenburg Sahlgrenska Academy, Clinical neuroscience and Rehabilitation
REVIEW RETURNED	10-Mar-2023

GENERAL COMMENTS	Comments to the authors: Thank you for giving me the possibility to review this manuscript about the patients experiences of undergoing assessment of cognitive function, a little explored area. I must say that - as a clinician- I'm not surprised by the findings and I believe the described patient reactions are common to meet in everyday clinical practise. I find the manuscript to be well written bur also quite wordy, although I think is acceptable when presenting research based on qualitative analysis. However, the authors are recommended to try to make the text more stringent. Introduction In general, I find the Introduction short and clear. However, I lack a description of the clinical perspective, that is: Although you write that cognitive assessment post-stroke are increasingly used in different clinical settings, I would like to know more about who administers these screens and how the results are used at present i.e. do they lead to more extensive neuropsychological assessments and/or to the patient being offered rehabilitation? P 5 Methods Patient and caregiver sampling Line 38: Can you give a more accurate description of the clinical setting at the John Radcliffe Hospital (acute setting? Rehabilitation services etc) Line 41 They were subsequently followed-up approximately 6 months later. Were the research interviews performed at this occasion? Line 54 patients who had provided opt-in consent for further research Line 57 Purposive sampling was used to ensure a mix of ages and sexes among participants. I think this is fair but would have liked a more explicit description of this. Were there some sort of criteria or recruitment goals, for
--

	instance concerning the proportions of men and women in the sample or the severity of the stroke? P 6 Line 3:stroke survivors with and without cognitive impairment.... Cognitive impairment according to what definition? Do you mean according to the OCS assessment or some other instrument? Does this refer to an early assessment or is it at the time of the interview? P 6 Line 35: Procedure Line 40: based on relevant literature.... Can you give some examples of “relevant literature” ? P 8 Line 15: The OCS was performed by occupational therapists..... Did the occupational therapist have any special training or education in administration cognitive tests? Line 22: Concerning the OCS Can you specify how the OCS score should be interpreted? (i.e a higher score means ...) I also think you should mention the NIHSS in this section. Page 8 Line 27 Data Analysis Line 47:including the analytic goals of the study.... What do you mean by “analytic goals”? Can you specify? Page 9 Line 11 Results No comments P 28 of 35 Line 34 Discussion I believe the Discussion mainly repeats what is already written in the Method and Results sections. A suggestion is to shorten the first section by describing your findings in just a few introductory sentences, which you then expand in the rest of the discussion.
--	---

REVIEWER	Langhammer, Birgitta Oslo Metropolitan University, Department of Physiotherapy
REVIEW RETURNED	13-Apr-2023

GENERAL COMMENTS	This is an important topic and I think you have pin pointed the sensitive area cognitive tests /screenings represent for the stroke population which may be of value for clinicians.
--

VERSION 1 – AUTHOR RESPONSE

Comments from Reviewer 1 (Dr. Caisa Hofgren, University of Gothenburg Sahlgrenska Academy, University of Gothenburg Institute of Neuroscience and Physiology)

Thank you for giving me the possibility to review this manuscript about the patients experiences of undergoing assessment of cognitive function, a little explored area. I must say that - as a clinician- I'm not surprised by the findings and I believe the described patient reactions are common to meet in everyday clinical practise.

I find the manuscript to be well written bur also quite wordy, although I think is acceptable when presenting research based on qualitative analysis. However, the authors are recommended to try to make the text more stringent.

Thank you for your kind review of our manuscript. We are pleased that Reviewer 1 has experienced similar views in everyday clinical practice. Qualitative research is often very challenging to condense, given the richness of the data and the lengthiness of participant quotes but we have condensed the manuscript where possible and provide examples below. We have also made our text more stringent by combining two themes which, on reflection, we felt were potentially overlapping. The revised manuscript word count is 5,448, compared to the original 6,030.

Abstract (Results):

- We identified three key phases of the cognitive assessment process and themes pertaining to each phase. The phases (numbered) and themes (lettered) were: (1) Before the cognitive assessment: (a) Lack of explanation, (b) Considering the assessment useless; (2) During the cognitive assessment: Varied emotional responses, moderated by (d) Perception of the purpose behind cognitive assessment, (e) Perception of cognitive impairment, (f) Confidence in cognitive abilities, (g) Assessment administration style; and (3) After the cognitive assessment: (h) Feedback can impact self-confidence and self-efficacy, (i) Vague feedback and clinical jargon are unhelpful.

Introduction:

- Stroke is among the most common causes of disability worldwide and nearly all stroke patients experience some level of cognitive impairment in the first weeks after stroke.(1–3) Acute cognitive impairment has implications for post-stroke recovery, by increasing the risk of post-stroke depression (4,5) and reducing quality of life.(6)

Methods (Participant Sampling):

- GH initially contacted sampled participants by telephone to provide them with a brief description of the study and offer them the opportunity to ask any questions. They were then sent a detailed information sheet and participant consent form to read. They were asked to phone the researcher to ask any questions and to organise a date for the semi-structured interview, should they decide to participate.

Methods (Procedure):

- GH conducted interviews between May 2022 and September 2022. Semi-structured interviews were conducted either at participants' homes or via telephone or videocall, with interview location determined by participants' individual preferences. Most interviews were conducted individually (i.e., interviewer and one participant). However, they were conducted in pairs (i.e., interviewer and two participants) if both the stroke survivor and their caregiver preferred to take part in the interview together. Interviews were audio-recorded with participants' consent. GH transcribed audio-recordings verbatim using Jeffersonian Lite style.(16) All identifiable personal data were removed from the transcripts and transcripts were labelled using unique participant identifiers.

Demographic and clinical data were retrieved from previous research data. Clinical data included NIHSS score, a measure of stroke severity recorded by clinical staff during acute hospital admission (14), and performance on the OCS. The OCS was administered by occupational therapists as part of routine care during hospitalisation for stroke using the procedure outlined by the OCS manual (8). Cognitive impairment classifications were determined using normative data from neurotypical adults. OCS administration, scoring, and classification of impairment is described in detail elsewhere.(8)

Results (Feedback can impact self-confidence and self-efficacy)

- One carer referred to her husband's self-critical tendencies after the stroke and speculated that professional feedback highlighting his remaining cognitive strengths could have reminded him of his intact abilities and increase his self-efficacy, but she was disappointed this had not happened in practice. Instead, she described how the small amount of feedback provided was biased towards the patient's weaknesses, rather than encapsulating areas in which he had performed well.

Discussion:

- The present study used a qualitative approach to explore experiences of the cognitive assessment process from the perspective of stroke survivors and their family members. Whilst recommended by clinical guidelines, post-stroke cognitive assessments had the potential to evoke strong emotional and psychological responses, which should be handled carefully by healthcare professionals. The present study highlights the potential benefit of providing a clear explanation of the purpose behind the cognitive assessment and constructive feedback afterwards. The development of clear guidelines and recommendations may help to standardise this process.

Introduction

In general, I find the Introduction short and clear. However, I lack a description of the clinical perspective, that is: Although you write that cognitive assessment post-stroke are increasingly used in different clinical settings, I would like to know more about who administers these screens and how the results are used at present i.e. do they lead to more extensive neuropsychological assessments and/or to the patient being offered rehabilitation?

Thank you. We now provide further information about this in the revised manuscript, referencing an important paper that comments on these questions (Ablewhite et al., 2019). We copy the relevant text below.

Introduction:

- Cognitive assessment tools designed specifically for stroke populations (e.g., Oxford Cognitive Screen)(8) are increasingly used for this purpose, as well as in community stroke settings.(9) Stroke-specific tools, such as the Oxford Cognitive Screen (OCS), are usually administered by occupational therapists as a means of first-line screening for cognitive problems after stroke and scores from these assessments are used to inform and plan rehabilitation programmes,(9) which may include further referral to clinical neuropsychology services where appropriate.

Methods

Patient and caregiver sampling: Can you give a more accurate description of the clinical setting at the John Radcliffe Hospital (acute setting? Rehabilitation services etc).

We have clarified this in the revised manuscript and copy the relevant text below.

Abstract (Setting):

- Stroke survivors were originally recruited from the acute inpatient unit at Oxford University Hospital (John Radcliffe), United Kingdom.

Methods (Participant Sampling):

- Stroke survivors were recruited through a pool of research volunteers who had previously taken part in the OCS-Recovery study (NHS REC reference 18/SC/05501). The OCS-Recovery study originally recruited a sample of stroke survivors from the acute stroke inpatient unit at Oxford University Hospital (John Radcliffe), United Kingdom.

“They were subsequently followed-up approximately 6 months later.” Were the research interviews performed at this occasion?

We now clarify that the research interviews were not performed during the follow-up stage. We also clarify that follow-ups were conducted several months later, with the cohort average follow-up time approximating six months after stroke. Research interviews were conducted with participants who had already completed their follow-up assessment and who had given opt-in consent to being contacted for further research opportunities. Therefore, research interviews took place after and separate from the OCS-Recovery follow-up assessments. We have clarified this in the revised manuscript and copy relevant text below.

Methods (Participant Sampling):

- The OCS-Recovery study originally recruited a sample of stroke survivors from the acute stroke inpatient unit at Oxford University Hospital (John Radcliffe), United Kingdom. Stroke survivors recruited to the OCS-Recovery study were visited for a follow-up assessment at their home after discharge. The average (mean) time since stroke at follow-up for the OCS-Recovery participant sample was approximately six-months. Inclusion criteria for the OCS-Recovery study were: at least 18 years old, clinical diagnosis of stroke, and ability to remain alert for at least 20 minutes at the point of recruitment. Patients provided written or witnessed informed consent at recruitment and at follow-up.

The present study iteratively sampled stroke survivors who had already completed their follow-up visit and provided opt-in consent to be contacted about further research participation.

“Purposive sampling was used to ensure a mix of ages and sexes among participants.”

I think this is fair but would have liked a more explicit description of this. Were there some sort of criteria or recruitment goals, for instance concerning the proportions of men and women in the sample or the severity of the stroke?

Thank you for this comment. We now explain our approach below and have provided additional detail in the revised manuscript.

We did not set rigid a priori goals for participant recruitment. That is, we did not precisely define the proportion of stroke survivors that should meet relevant demographic or clinical criteria. Instead, we considered that stroke survivors with different characteristics – sex, stroke severity (NIHSS), and cognitive impairment severity (OCS performance) – would likely hold different yet important views pertaining to our research question. Therefore, we aimed to include in our final participant sample: (a) both male and female stroke survivors; (b) stroke survivors with mild (0-5), moderate (5-14), and severe (15-24) NIHSS scores; and (c) both stroke survivors with no cognitive impairments during acute hospitalisation and those with cognitive impairments affecting different cognitive domains (i.e., language, memory, attention, executive function, number processing, and praxis).

Methods (Participant Sampling):

- The research team considered that stroke survivors with different characteristics, including sex, level of stroke severity, and types of domain-specific cognitive impairment, would likely hold different but important views pertaining to the main research question. Therefore, purposive sampling was used to ensure the sample included both male and female stroke survivors, stroke survivors with mild, moderate, and severe stroke, and stroke survivors with domain-specific cognitive impairments affecting different domains. Stroke severity was assessed using the National Institute of Health Stroke Scale (NIHSS) score, which was recorded during acute hospital admission.(14) Cognitive impairment was determined based on performance in the OCS relative to normative cut-offs during acute hospital admission.

“...stroke survivors with and without cognitive impairment...”

Cognitive impairment according to what definition? Do you mean according to the OCS assessment or some other instrument? Does this refer to an early assessment or is it at the time of the interview?

We now clarify this and have made this explicit:

Methods (Participant Sampling):

- Cognitive impairment was determined based on performance in the OCS relative to normative cut-offs during acute hospital admission.

“.... based on relevant literature....” Can you give some examples of “relevant literature” ?

Thank you for this suggestion. We now give examples of articles used to inform the development of our topic guide for the semi-structured interviews:

Methods (Procedure):

- The research team developed a topic guide for the semi-structured interviews based on relevant literature (e.g., 12,13) and their clinical expertise.

“The OCS was performed by occupational therapists.....” Did the occupational therapist have any special training or education in administration cognitive tests?

The occupational therapists referred to in our manuscript were employed by Oxford University Hospital and therefore followed clinical training procedures as part of their professional training and as part of being employed by the relevant NHS Trust. This involved shadowing of cognitive assessment administration, followed by supervised administration, prior to independent administration. We note that the OCS manual provides extensive detail about how the OCS should be administered and scored.

Methods (Procedure):

- The OCS was administered by occupational therapists as part of routine care during hospitalisation for stroke using the procedure outlined by the OCS manual (see 8 for further detail).

Concerning the OCS, can you specify how the OCS score should be interpreted? (i.e a higher score means ...)?

It is not straightforward to specify how the OCS score should be interpreted as a whole, as each task is scored separately, to reflect domain-specific cognition. Nevertheless, in order to provide a simple overview of severity, an overall number of OCS tasks impaired score is provided in Table 1. A higher score here means more cognitive tasks were impaired on the OCS. There is a detailed description of OCS scoring and interpretation provided by Demeyere et al. (2015), which is referenced in addition.

- Table 1. Demographic and clinical data for stroke survivors interviewed for the present study. Acute OCS refers to performance on the Oxford Cognitive Screen during acute hospital admission. We report the average number of tasks on which stroke survivors were categorised as impaired, relative to normative data, with higher scores denoting poorer performance (i.e., more task impairments). The OCS includes a total of 12 tasks.

Methods (Procedure):

- The OCS was administered by occupational therapists as part of routine care during hospitalisation for stroke using the procedure outlined by the OCS manual.(8) Cognitive impairment classifications were determined using normative data from neurotypical adults. OCS administration, scoring, and classification of impairment is described in detail elsewhere.(8)

I also think you should mention the NIHSS in this section.

We agree and have revised our manuscript accordingly.

Methods (Procedure):

- Demographic and clinical data were retrieved from patients' previous research data. Clinical data included NIHSS score, a measure of stroke severity recorded by clinical staff during acute hospital admission, (14) and performance on the OCS.

".....including the analytic goals of the study...." What do you mean by "analytic goals"? Can you specify?

Our primary analytic goal was to answer the question "what are the experiences of stroke survivors completing cognitive assessments?" We have now clarified this explicitly:

Methods (Data Analysis):

- The decision to cease data collection and analysis was based on several factors, including the analytic goals of the study (i.e., to develop a theme structure encapsulating experiences of cognitive assessment after stroke) and pragmatic constraints on time and resources.(19)

I believe the Discussion mainly repeats what is already written in the Method and Results sections. A suggestion is to shorten the first section by describing your findings in just a few introductory sentences, which you then expand in the rest of the discussion.

We have made our discussion more concise by describing our findings very briefly initially and then expanding upon them. We have also condensed other sections in the discussion (please see below examples) and we have integrated the clinical implications of our findings into the Discussion more thoroughly so that it clearly expands upon the results reported in the Methods.

Discussion:

- The present study used a qualitative approach to explore experiences of the cognitive assessment process from the perspective of stroke survivors and their family members. Whilst recommended by clinical guidelines, post-stroke cognitive assessments had the potential to evoke strong emotional and psychological responses, which should be handled carefully by healthcare professionals. The present study highlights the potential benefit of providing a clear explanation of the purpose behind the cognitive assessment and constructive feedback afterwards. The development of clear guidelines and recommendations may help to standardise this process.

- Stroke survivors reported receiving different amounts and types of feedback after the cognitive assessment, indicating a potential lack of consistency in the feedback provided in clinical practice. This highlights a need for clearer guidelines to standardise what and how information is communicated to stroke survivors and their caregivers after cognitive assessments. Clinical guidelines should consider the emotional impact of cognitive assessment feedback, as the present study revealed the potential for feedback to improve confidence and self-efficacy, both of which have been identified as important factors in promoting post-stroke recovery.(23,24) Our interviews also demonstrated the importance of communicating feedback using lay terminology, echoing the results of previous research that looked at patient responses to jargon in other clinical populations.(25) However, further research is required to clarify precisely which words hinder patients' understanding and to identify acceptable alternatives.

Comments from Reviewer 2 (Dr. Birgitta Langhammer, Oslo Metropolitan University, Sunnaas Sykehus HF)

This is an important topic and I think you have pin pointed the sensitive area cognitive tests /screenings represent for the stroke population which may be of value for clinicians.

We would like to thank Reviewer 2 for their kind review of our manuscript. We are particularly pleased that Reviewer 2 recognises the potential value of our work for clinicians.

VERSION 2 – REVIEW

REVIEWER	Hofgren, Caisa University of Gothenburg Sahlgrenska Academy, Clinical neuroscience and Rehabilitation
REVIEW RETURNED	25-May-2023
GENERAL COMMENTS	After the revision I now find the manuscript much improved, in that it clarifies the questions I initially had and I suggest it now can be accepted for publication. Hopefully you will continue research within this field in order to improve information and treatment for patients concerning cognitive examinations.

VERSION 2 – AUTHOR RESPONSE